# Electronic and Nuclear Quantum Effects on Proton Transfer Reactions of Guanine–Thymine (G-T) Mispairs Using Combined Quantum Mechanical/Molecular Mechanical and Machine Learning Potentials

**DOI:** 10.3390/molecules29112703

**Published:** 2024-06-06

**Authors:** Yujun Tao, Timothy J. Giese, Darrin M. York

**Affiliations:** Laboratory for Biomolecular Simulation Research, Institute for Quantitative Biomedicine and Department of Chemistry and Chemical Biology, Rutgers University, Piscataway, NJ 08854, USA; yujun.tao@rutgers.edu (Y.T.); timothyjgiese@gmail.com (T.J.G.)

**Keywords:** DNA mispair, PIMD, machine learning, tautomerization

## Abstract

Rare tautomeric forms of nucleobases can lead to Watson–Crick-like (WC-like) mispairs in DNA, but the process of proton transfer is fast and difficult to detect experimentally. NMR studies show evidence for the existence of short-time WC-like guanine–thymine (G-T) mispairs; however, the mechanism of proton transfer and the degree to which nuclear quantum effects play a role are unclear. We use a B-DNA helix exhibiting a wGT mispair as a model system to study tautomerization reactions. We perform *ab initio* (PBE0/6-31G*) quantum mechanical/molecular mechanical (QM/MM) simulations to examine the free energy surface for tautomerization. We demonstrate that while the *ab initio* QM/MM simulations are accurate, considerable sampling is required to achieve high precision in the free energy barriers. To address this problem, we develop a QM/MM machine learning potential correction (QM/MM-ΔMLP) that is able to improve the computational efficiency, greatly extend the accessible time scales of the simulations, and enable practical application of path integral molecular dynamics to examine nuclear quantum effects. We find that the inclusion of nuclear quantum effects has only a modest effect on the mechanistic pathway but leads to a considerable lowering of the free energy barrier for the GT*⇌G*T equilibrium. Our results enable a rationalization of observed experimental data and the prediction of populations of rare tautomeric forms of nucleobases and rates of their interconversion in B-DNA.

## 1. Introduction

With five natural nucleobase monomers (G, C, A, and T/U), DNA and RNA are able to store and transfer a rich density of genetic information with high fidelity [1,2]. Nonetheless, the diversity of biological populations arises from processes of natural selection and genetic variation [3,4], the latter of which is influenced by mutations that occur during DNA replication [5,6]. Even with the precise replication process [7], including proofreading [8] and DNA repair [8], the mutation rate still typically reaches $10^{-9}∼10^{-12}$ per base pair synthesized [9]. While mutations are a vital component in the creation of genetic diversity, some mutations in eukaryotes lead to diseases, including many forms of cancer [10,11]. Mutations can arise through mispairing of the Watson–Crick G-C and A-T hydrogen bonding bases in the DNA helix. In the Watson–Crick (WC) model, nucleobase pairs are in their “keto” form [12] rather than their “imino” or “enol” forms. Mispairs are believed to commonly arise from tautomeric shifts in the nucleobases [5,6] that cause a change in the hydrogen bond pattern at the Watson–Crick edge without changing the charge [2]. These rare nucleobase tautomers are able to form WC-like mispairs that can sometimes circumvent proofreading and repair machinery and lead to mutations and misincorporation in DNA replication [7,9] and translation [13]. Hence, the study of how protons transfer between different tautomeric forms in DNA mispairs is of fundamental importance.

Despite their importance, the experimental detection of tautomeric nucleobases is challenging due to their short lifetimes [2,14]. Nonetheless, much insight has been gained by previous experimental studies. Allawi *et al.* investigated the stability of 39 oligonucleotides with internal G-T mismatch using UV absorbance versus temperature profiles and NMR [15]. Waters *et al.* reported the kinetics of removal of thymine DNA glycosylase from DNA containing a G-T mismatch [16]. Fox *et al.* reported a G-T mismatch mechanism by using base analogues of G and T [17] to explore the functional groups on the mismatch pair which are recognized by the enzyme. Osakada *et al.* studied the kinetics of charge transfer for the detection of single-base mismatch in a DNA molecule [18]. Koag *et al.* studied mismatch discrimination mechanisms in human DNA polymerase β crystal structures and proposed a two-stage mechanism where in the closed conformation state, polβ allows only a Watson–Crick-like conformation [19].

In pioneering NMR work, Al-Hashimi and coworkers reported the existence of transient WC-like G-T mispairs, providing insight into their populations and rates and establishing a dynamical basis for misincorporation via tautomerization and, more rarely, ionization [20,21,22]. In a follow-up computational study [23] focused on calculation of the relative tautomer free energies, it was shown that the tautomer equilibrium was sensitive to the B-DNA environment. This equilibrium was predicted to be shifted to become more favorable when bound to DNA polymerase λ, as was proposed in earlier work [24].

Other computational studies have examined tautomerization in DNA [5] and G-U wobbles in RNA [25]. Very recently, tautomerization in GT wobble pairs has been studied using a coupled first-principles quantum chemistry and open quantum system master equation approach [26]. Results illustrate that quantum tunneling plays an important role in these reactions and has significant influence on rates and transcription error frequency.

In the present work, we examine the mechanism of tautomerization of G-T mispairs in B-DNA, with emphasis on the reaction paths and transition states, sensitivity of the free energy profiles and barriers to sampling, and inclusion of nuclear quantum effects. As a model system, we use the B-DNA crystal structure [27] with two G-T wobble pair mismatches, illustrated in Figure 1. We perform a series of combined quantum mechanical/molecular mechanical (QM/MM) simulations using semiempirical, *ab initio*, and machine learning models to study the mechanism of the tautomerization reaction. Path integral molecular dynamics (PIMD) simulations are performed to deduce the role of nuclear quantum effects on the calculated barriers in order to help rationalize the observed experimental data. The remainder of the manuscript is outlined as follows. The Section 3 provides a description of the computational details, including the QM/MM and QM/MM-ΔMLP models, construction of the free energy profiles, and PIMD methods. The Section 2 compares free energy profiles using different QM/MM and QM/MM-ΔMLP models, examines the sensitivity of the profiles to degree of sampling, and predicts the influence of nuclear quantum effects on the barriers. Results are discussed in the context of experiments and other related computational work reported in the literature. This paper concludes with a summary of key take-home points.

## 2. Results and Discussion

Figure 1 shows the B-DNA sequence and structure studied in the current work. The figure highlights the G-T wobble (wGT), and it illustrates the general process of tautomerization. The formation of rare GT* and G*T states, where G* and T* indicate the higher-energy enol tautomeric form of G and T, respectively, leads to Watson–Crick-like mispairs containing three hydrogen bonds, similar to a GC base pair. The wGT mispairs have been proposed to lead to misincorporation errors in DNA replication [21,23]. Tautomerization in GT mispairs have been studied previously using *ab initio* QM/MM simulations [23], and it has been shown that the tautomer equilibrium is sensitive to the environment. Specifically, Li *et al.* reported an insightful study of the reaction in aqueous solution in canonical A-form and B-form DNA helices in solution, as well as B-DNA bound to DNA polymerase λ. The main goal of the work was to compute the relative free energies of the wGT, GT*, and G*T tautomers, with less emphasis on the reaction paths and barriers as is the focus of the current work. Results of *ab initio* QM/MM simulations predicted that the wGT→GT* tautomerization is endoergic in B-DNA in aqueous solution but exoergic in the polymerase environment [23].

The goal of the present work is to use a B-DNA helix exhibiting a wGT mispair as a model system to study the tautomerization reactions in greater detail with regard to the reaction paths and transition states, sensitivity of the free energy profiles and barriers to sampling, and inclusion of nuclear quantum effects. The following briefly highlights the main methodological differences in the classical *ab initio* QM/MM simulations used in the current work with respect to that of Li *et al.* [23]. The latter study used the ωB97X-D3 functional [28,29] and 6-311G** basis set [30,31] and performed string calculations using 25 images with 0.1 ps sampling per image. Simulations were performed under aperiodic (frozen) boundary conditions outside of an 18 Å distance. In the current work, the PBE0/6-31G* *ab initio* QM/MM model was used under full periodic boundary conditions (no frozen boundary) with rigorous electrostatics [32], and string simulations were conducted with 32 images, 2 ps sampling per image, and 40 ps of production sampling (four independent 10 ps trials) along the final path in order to accurately obtain barriers.

In order to ascertain the contribution of nuclear quantum effects, path integral simulations were performed with a recently developed software interface [33,34] between Amber [35] and i-PI [36]. PIMD simulations are roughly an order of magnitude more computationally costly than the classical MD simulations and preclude the practical use of the *ab initio* QM/MM model. To address this barrier, we developed a QM/MM-ΔMLP model based on a fast, semiempirical QM/MM model that was more than 100 times faster than the *ab initio* QM/MM model but with comparable accuracy.

The next section compares free energy profiles for tautomerization reactions from classical MD using semiempirical and *ab initio* QM/MM models as well as the QM/MM-ΔMLP developed in this work. The following section then applies the QM/MM-ΔMLP model to ascertain the effect of inclusion of nuclear quantum effects on the reaction barriers. The final section ties the results together and compares the various models with other computational results reported in the literature as well as available experiments.

### 2.1. Free Energy Profiles of G-T Mispair from Classical Molecular Dynamics

Figure 2 compares the free energy profiles from classical MD simulations for the *ab initio* QM/MM (PBE0/6-31G*), semiempirical QM/MM (AM1/d), and QM/MM-ΔMLP models. Table 1 lists the relative free energy values for key stationary points along the various profiles. The reactions are broken down into competing wGT→GT* and wGT→G*T pathways and, separately, the interconversion GT*→G*T. These reaction steps are shown schematically in Figure 3, and representative structures for key stationary points along the reaction paths are shown in Figure 4. The 1D profiles shown are the result of the minimum free energy path derived from the finite temperature string method [37,38], followed by additional production path refinement sampling (see Section 3).

The *ab initio* QM/MM profiles have a single transition state with the exception of the wGT→G*T profile. The wGT→GT* profile passes through a single transition state (TSa) that involves a partially ionized state with formal charges at G:O6 (protonated) and T:O4 positions (Figure 3a) that then undergoes a conformational shearing of the base pair concerted with proton transfer to form GT*. The wGT→GT* profile follows an initial path (Figure 3b) to a partially ionized transition state (TSb1) similar to that of TSa, but then follows a stepwise path that first involves conformational shearing of the ionized nucleobases to form a metastable intermediate (Ib) that then undergoes a proton transfer from G:N1→T:N3 passing through transition state (TSb2) to form the GT* product. The GT*→G*T involves a concerted dual proton transfer (Figure 3b) that passes through a low-barrier charge-separated transition state (TSc).

The AM1/d QM/MM method is a fast, semiempirical model that was first tested to ascertain the degree to which it could reliably model tautomerization reactions such that it might be applied with the more computationally intensive PIMD simulations. The AM1/d method considerably overestimates the free energy barriers relative to the *ab initio* QM/MM model (Figure 2). Additionally, the AM1/d model produces wGT→GT* and wGT→G*T reaction profiles that are more endoergic. This is likely due in part to the known problem that AM1 tends to underpredict the strength of hydrogen bonds. As the formation of the high-energy enol tautomer states (G*T and GT*) is accompanied by an increase in the number of hydrogen bonds, an underprediction of the strength of the hydrogen bonds will lead to a higher-energy Watson–Crick-like hydrogen bonded state. Perhaps more concerning is that the AM1/d profile produces multiple transition states for each profile separated by, in some cases, artificial intermediates not observed in the *ab initio* QM/MM profiles.

The developed QM/MM-ΔMLP model, on the other hand, is observed to closely mimic the *ab initio* QM/MM profiles in Figure 2. The transition state barriers ($\Delta A^{‡}$) are all within approximately 1 kcal/mol of the PBE0/6-31G* QM/MM values (Table 1), and the reaction free energy values (ΔA) are less than 0.5 kcal/mol. This provides support for the use of the QM/MM-ΔMLP model in the PIMD simulations described in the Section 2.3.

### 2.2. Sensitivity of Free Energy Profiles to Sampling

The quantitative determination of free energy surfaces and minimum free energy paths for complex systems requires intensive sampling efforts. Oftentimes, even a minimal level of sampling is precluded by the computational cost of *ab initio* QM/MM simulations, particularly when the QM region becomes large. In this section, we illustrate the degree to which sampling can affect predictions obtained from the free energy profiles examined in this study.

In the current work, we performed finite temperature string method optimization using five generalized coordinates to determine the minimum free energy path for the reaction. At the *ab initio* QM/MM level, we then refined the final path with four independent trials, each of which were conducted for 10 ps for each of the 32 string images (windows). The cumulative sampling from the string simulations and final path refinement were used to construct the plots shown in Figure 2. For the more efficient QM/MM-ΔMLP, the refinement was conducted with four independent trials, and each of the 32 images was sampled for 25 ps/image/trial. This is considerably more sampling than what is often encountered in other studies. For example, a previous study of the same reaction used approximately 2.5 ps/image of cumulative sampling to construct free energy profiles [23].

In order to make an estimate of the expected variance of the free energy profiles derived from more modest sampling, we divided the 40 ps of sampling per string image into 16 nonoverlapping 2.5 ps blocks. We then reanalyzed the free energy profiles using only the 2.5 ps of sampling of a single block in order to come up with 16 profiles, which are shown in Figure 5. While the overall shape of the profiles is qualitatively similar, the range of values for the rate-controlling transition states is observed to vary between 2.0 and 3.4 kcal/mol, particularly for the GT*→GT tautomerization. One should bear in mind that this demonstration is a best-case scenario in the sense that it examines the refinement of a final converged path. It does not consider variations in the path that could result from independent string simulations that also used less sampling to obtain the final path. This demonstration highlights the need for sampling in order to obtain sufficiently precise quantitative estimates of the free energy profiles and barriers. In turn, this stresses the importance of developing accurate QM/MM-ΔMLP models that are able to access time scales well beyond that of conventional *ab initio* QM/MM methods. This becomes even more critical when one considers more computationally intensive PIMD simulations in the Section 2.3.

### 2.3. Free Energy Profiles of G-T Mispair from Path Integral Molecular Dynamics

Figure 6 compares the classical and PIMD free energy profiles using the QM/MM-ΔMLP model. The differences in free energy values between classical MD and PIMD, as well as approximate changes in rates and populations, are summarized in the Appendix A. For the wGT→GT* and wGT→G*T reactions, the profiles and barriers are similar. This is due to the nature of the rate-controlling transition state that involves a conformational event (base shearing, as illustrated in Figure 3 and Figure 4, TSa and TSb1), as opposed to a proton transfer event. Alternatively, in the case of the GT*→G*T, it is the proton transfer itself (TSc in Figure 3 and Figure 4) that characterizes the rate-controlling transition state, and the barrier is predicted to be significantly lower with PIMD.

Generally, the inclusion of nuclear quantum effects does not significantly alter the reaction free energies for the systems examined here. This implies that the populations of the tautomer states are predicted by classical and PIMD simulations to be very similar. The most change in populations is for GT*→G*T, where the ΔA value for the reaction changes from −0.43 to 0.26 kcal/mol. The concerted proton transfer involved in the equilibrium between GT* and G*T is predicted to be more rapid when nuclear quantum effects are included. A rough estimate based only on the $\Delta \Delta A^{‡}$ values suggests that PIMD would increase the forward rate by a factor of 123 and the reverse rate by a factor of 395.

The evolution of the key distances as a function of the reaction path progress variable [0,1], as well as the proton ring polymer degree of expansion ($\Delta |r_{H}|$, Equation (Equation 1)) that is a measure of the delocalization of the proton wave packet [40], is shown in Figure 7. In each case, pronounced delocalization of the proton wave packet occurs at the crossing point of covalent bond formation and cleavage distances involved in the proton transfer. In the wGT→GT* and wGT→G*T reactions, the proton transfers are distinctly separated, whereas in the GT*→G*T reaction, they are concerted and appear in close proximity in terms of the path progress variable. Similar behavior has been observed for proton transfer reactions elsewhere [40].

### 2.4. Comparison of Various Models with Experiment

Table 1 compares the free energy values for each of the models with values reported by Li *et al.* [23] and from the experiment. Differences in free energies with respect to the experiment are listed as Δ values. Overall, the free energy differences ΔA between the GT wobble pair and the GT* and G*T states are predicted from calculations to be too endoergic. The AM1/d model performs the worst, with ΔA values exceeding 8 kcal/mol, whereas the PBE0/6-31G* QM/MM and QM/MM-ΔMLP models have errors below 4 kcal/mol. For the wGT→GT* reaction, Li *et al.* [23] reported a value of 6.00 kcal/mol, which is in good agreement with the experimentally derived value of 4.43 kcal/mol (estimated from the measured rate using classical transition state theory with a unit transmission coefficient). The free energy barriers for the wGT→GT* reaction are overestimated using PBE0/6-31G* QM/MM and QM/MM-ΔMLP, while they appear underestimated for GT*→G*T.

The general picture that emerges from the current work together with previous computational [23,34] and experimental [20,21,22] work is that for the B-DNA system, there is a relatively high barrier for the tautomerization of the wGT wobble pair to form either GT* or G*T tautomer states. The barrier for interconversion between GT*→G*T is much lower, and this barrier is predicted to be considerably influenced by nuclear quantum effects. Likely, this implies that there is a rapid equilibrium between GT* and G*T states that occurs on a time scale much faster than the rate of transition from wGT.

In a broader context, as alternative tautomeric forms of DNA bases have been implicated in DNA replication, it is of interest to consider how tautomerization is affected by the environment of a DNA polymerase. In the study of Li *et al.* [23], the GT*→G*T tautomerization was studied in B-DNA and also in a DNA polymerase λ variant. The study found that the environment of the polymerase had a significant influence on the thermodynamics, kinetics, and mechanism. Further, Slocombe *et al.* examined quantum tunneling effects for tautomerization in GT wobble pairs by coupling first-principles quantum chemistry calculations with an open quantum system master equation [26]. Results predicted the existence of a transient “tunneling-ready” state in the polymerase active site that led to a 100-fold increase in rate. In the present work, simulations were not performed in the polymerase environment. Nonetheless, given that the environment alters the population of G-T mispairs and the transition state ensemble for tautomerization [23], together with evidence for a “tunneling-ready” state in the polymerase [26], it is likely that polymerase binding would impact the nuclear quantum effects. The degree to which nuclear quantum effects are influenced by different environments is a subject of continued interest and future work.

### 2.5. Benefits and Pitfalls of the QM/MM+ΔMLP Approach

The approach taken in the current work is to train a QM/MM+ΔMLP model to closely reproduce *ab initio* QM/MM simulation data at the PBE0/6-31G* level in order to accelerate path sampling convergence and enable more practical PIMD simulations. The advantage of such an approach is that once developed, the QM/MM+ΔMLP model affords a factor of roughly 300-fold speedup relative to (PBE0/6-31G*)/MM simulations with an accuracy of free energy values within 1 kcal/mol (Table 1). The disadvantage of this approach is that one must train the QM/MM+ΔMLP model itself, and this will depend on the training data. In the present case, we train against (PBE0/6-31G*)/MM energy and force data. The reliability of these data will depend not only on the level of quantum theory (e.g., the chosen exchange–correlation functional and basis set) but also on the QM/MM interaction parameters (i.e., the MM charges and Lennard–Jones parameters). Recently, the ωB97M-D3BJ/def2-TZVPPD level of theory has been demonstrated to be highly reliable and has been used to create databases for intermolecular interactions [41] that in turn have been used to create machine learning potentials [42] for drug discovery. Such an approach, however, becomes more difficult when transferred to the study of complex chemical reactions that occur in heterogeneous condensed-phase environments such as those considered in the current work. An additional concern when applying a QM/MM+ΔMLP model is that it may not reliably represent the target *ab initio* QM/MM model outside the scope of the training data ensemble. One possible mechanism to perform a consistency check is to use a weighted thermodynamic perturbation approach [43] or generalized multireference variant [44]. These methods potentially can correct the approximate QM/MM+ΔMLP free energy surface to the target *ab initio* QM/MM level, provided that there is sufficient phase space overlap as indicated by an analysis of the reweighting entropies.

**Table 1 molecules-29-02703-t001:** Free energy values along the reaction path for tautomerization reactions using different QM models and methods ^*a*^.

Reaction	Method	$\Delta A(\sigma_{\mathrm{err}})$	Δ	$\Delta A_{f}^{‡}\left(\sigma_{\mathrm{err}}\right)$	Δ	$\Delta A_{r}^{‡}\left(\sigma_{\mathrm{err}}\right)$	Δ
wGT→GT*	Expt. [21]	4.43		16.88		12.45	
	PBE0/6-31G*	3.97 (0.04)	−0.46	20.41 (0.05)	3.53	16.45 (0.04)	4.00
	AM1/d	9.54 (0.04)	5.11	23.51 (0.06)	6.63	13.98 (0.05)	1.53
	QM/MM-ΔMLP	3.6 7(0.07)	−0.76	21.01 (0.07)	4.13	17.34 (0.07)	4.89
	PIMD	3.48 (0.03)	−0.95	20.65 (0.04)	3.77	17.16 (0.03)	4.71
	Li *et al.* [23]	6.00	1.57	15.70	−1.18	9.70	−2.75
wGT→G*T	Expt. [21]	3.82					
	PBE0/6-31G*	3.06 (0.04)	−0.76	20.08 (0.05)		17.02 (0.05)	
	AM1/d	8.19 (0.04)	4.37	23.55 (0.04)		15.36 (0.04)	
	QM/MM-ΔMLP	3.24 (0.07)	−0.58	21.02 (0.07)		17.78(0.07)	
	PIMD	3.75 (0.04)	−0.07	20.63 (0.04)		16.88 (0.04)	
	Li *et al.* [23]	N/A		N/A		N/A	
GT*→G*T	Expt. [21]	−0.62		9.21		9.83	
	PBE0/6-31G*	−0.91 (0.03)	−0.29	6.72 (0.05)	−2.49	7.63 (0.05)	−2.20
	AM1/d	−1.36 (0.03)	−0.74	10.26 (0.03)	1.05	11.62 (0.04)	1.79
	QM/MM-ΔMLP	−0.43 (0.07)	0.19	7.06 (0.07)	−2.15	7.49 (0.07)	−2.34
	PIMD	0.26 (0.03)	0.88	4.21 (0.03)	−5.00	3.95 (0.04)	−5.88
	Li *et al.* [23]	−0.20	0.42	5.70	−3.51	5.90	−3.93

^*a*^ $\sigma_{\mathrm{err}}$ refers to the standard error of the mean for 4 trial runs; Δ refers to the difference between the current method and experimental results. Experimental results are obtained from Ref. [21]; Li *et al.* results were obtained in Ref. [23], which were calculated with ωB97X-D3/6-311G** and 13 reaction coordinates for a B-DNA system built by NAB program in AmberTools [45].

## 3. Materials and Methods

### 3.1. Background

There have been a number of quantum mechanical studies that have investigated the tautomerization of nucleobases in DNA and RNA [2,6,25,46,47,48,49,50,51] as well as extended synthetic genetic alphabets [2] that have great promise in the development of new technology. There have been fewer computational studies that utilize QM/MM simulations [23,52] to characterize the tautomerization reaction free energy path in complex condensed-phase environments. Further, there have been relatively few studies that have explored the contribution of nuclear quantum effects on the proton transfers involved in the tautomerization of nucleic acids [26,52,53,54].

One strategy that has been used to study nuclear quantum effects (within the Born–Oppenheimer approximation) in condensed-phase simulations of biological systems has been to use path integral molecular dynamics (PIMD) simulation methods [55,56,57,58,59]. Kosugi *et al.* [40] calculated among the first multidimensional quantum free energy surfaces using PIMD and a DFTB QM/MM model of a proton transfer reaction in a 2,4-dichlorophenol-trimethylamine complex. The QM/MM model’s empirical repulsive potential was parametrized to reproduce high-level QM reference data. The calculations used the multidimensional blue moon ensemble method to achieve a converged free energy surface in a tour de force benchmark demonstration. Sauceda *et al.* [60] more recently provided evidence that nuclear quantum effects enhance electronic interactions, leading to strengthening of covalent and noncovalent molecular interactions at finite temperature. The study used the symmetric gradient-domain machine learning (sGDML) framework [61,62,63] trained against high-level data sets [64,65]. The inclusion of nuclear quantum effects with PIMD can become cost-prohibitive at the *ab initio* QM/MM level when the quantum region becomes too large and/or has too intensive sampling requirements [66,67]. An alternative strategy is to use a fast, approximate, semiempirical [68] or density-functional tight-binding [69] QM model within a QM/MM or so-called quantum mechanical force field [66,67,70] framework. These models can be parameterized to improve their accuracy, for example, using force matching to higher levels [71,72,73]. Machine learning potentials (MLPs) have shown particular promise in enhancing the accuracy and performance of condensed-phase simulations of chemical reactions [33,74,75,76,77,78]. Of particular relevance to the current work is the development of QM/MM-ΔMLP models, whereby the energies and forces of a fast, approximate QM model are corrected with a machine learning potential [76,79,80,81,82,83,84,85]. These are described in more detail in the Appendix A.

### 3.2. Free Energy Profiles of Tautomer Reactions from Classical Molecular Dynamics

The model system was prepared from the crystal structure of B-DNA with G-T wobble pair mismatches (PDB ID:113D) [27]. Experimental NMR data for the G-T mispair dynamics were previously reported for this system [20,21,22]. The model contains 762 solute atoms (including hydrogen atoms). We solvated the system with 5151 water molecules in a truncated octahedron with 59.3 Å real-space lattice vector lengths. A total of 13 Cl^−^ and 35 Na^+^ ions were added to neutralize the charge and produce a 0.14 M ion concentration. The system was equilibrated with an MM potential following the procedure described in Ref. [86]. The procedure includes geometry optimization, heating, solvent annealing, and equilibration of the system density while restraining solute heavy atoms. The solute restraints are gradually reduced over the course of the equilibration procedure. This procedure involved 6.2 ns and 2.6 ns of simulation in the NVT and NPT ensembles, respectively, using a 1 fs time step. The system was then equilibrated for an additional 100 ns in the NPT ensemble. A Langevin thermostat [87] was used at a 5.0 ps^−1^ collision frequency to maintain the temperature at 298 K, and the density was equilibrated at 1 atm using a Berendsen isotropic barostat.

The DNA was modeled with the OL15 force field [88]. The SPC/Fw water model [89] and the Li *et al.* ion parameters [90] were used to represent the solvent environment. The particle mesh Ewald method [91,92] was used to calculate electrostatic interactions with a 10 Å real-space cutoff, a 1 Å reciprocal space grid spacing, and tinfoil boundary conditions.

The tautomeric reactions were decomposed 3 steps. Each step was a path that connected 2 tautomeric forms. The free energy profiles of each step were calculated from QM/MM umbrella sampling. The “wGT” state corresponded to the G-T wobble base pair in which the T21:N3 position was protonated. The “GT*” state was a tautomer in which the T21:H3 proton was covalent-bonded to the T21:O4 position. Similarly, the “G*T” state was a tautomer in which the H3 proton was bound to the G4:O6. The 3 steps involved proton displacement between these states: wGT→GT*, wGT→G*T, GT*→G*T.

The free energy profiles were calculated with several QM/MM potentials. The QM/MM methods compared in this work are as follows: PBE0/6-31G*, AM1/d [93,94,95], and the QM/MM-ΔMLP method described above. All QM/MM simulations were performed with rigorous long-range electrostatic interactions under periodic boundary conditions using linear-scaling QM/MM Ewald methods [32,96]. The QM region consisted of 51 atoms (the nucleobase and sugar of G4 and T21), and it had a net neutral charge. The minimum free energy path of each step was optimized in the space of 5 reaction coordinates with the finite temperature string method [38] for every QM/MM model. The 5 reaction coordinates were distance differences meant to represent the transfer of H3 and H1 protons and the relative displacement of the hydrogen bond pattern: $\xi_{1}=R_{N3-H3}-R_{O6-H3}$, $\xi_{2}=R_{O6-H3}-R_{O4-H3}$, $\xi_{3}=R_{N1-H1}-R_{N3-H1}$, $\xi_{4}=R_{N1-O2}-R_{N2-O2}$, $\xi_{5}=R_{N2-O2}-R_{N1-N3}$. The umbrella biasing potentials used 200 kcal mol^−1^ Å^−2^ force constants. The string method began from a linear interpolation of the reaction coordinates between the step’s reactant and product states. Each string was sampled with 32 umbrella windows, and the initial configurations were prepared in sequence starting from the reactant state. The QM/MM structure of each window was geometry-optimized for 1000 steps using the conjugate gradient method. The temperature was raised from 0 K to 298 K over the course of 40 ps, and 10 ps of QM/MM equilibration was performed in the NVT ensemble. The string method was performed for 50 iterations, and sampling occurred at 4 ps/window/iteration using 1 fs time step. Four sets of production simulations were performed along the optimized path using different thermostat random number seeds. Each production simulation included 25 ps/window of sampling, and the reaction coordinate values were recorded every 10 fs. The aggregate sampling from all 3 reaction steps was analyzed with the multistate Bennett’s acceptance ratio (MBAR) method [97], as implemented in the ndfes module in FE-ToolKit software [98,99], distributed within AmberTools [100], to produce 5-dimensional free energy surfaces. The profiles presented in this manuscript are the free energy values within the 5-dimensional surface evaluated along the minimum free energy path. PBE0/6-31G* simulations are very expensive; for this particular case, only 10 iterations of the string method were performed, starting from the optimized AM1/d path. The PBE0/6-31G* sampling was limited to 2 ps/window/iteration, and 4 sets of production simulations were run for 10 ps/window/trial with a 1 fs time step. The PBE0/6-31G* QM/MM electrostatics were calculated with the ambient-potential composite Ewald method [32] using a 10 Å real-space cutoff, 1 Å reciprocal space grid spacing, and tinfoil boundary conditions.

### 3.3. Free Energy Profiles of Tautomer Reactions from PIMD

We performed PIMD simulations using i-PI 2.0 software [36] to treat nuclear quantum effects. i-PI software [36] is a standalone molecular dynamics program that supports state-of-the-art path integral sampling [101], including the PIGLET thermostat, which has been shown to reduce the expense of computing quantum kinetic energy [102,103]. The path integral dynamics was performed with a ring polymer Hamiltonian consisting of several replicas (beads) that were harmonically restrained in series. At each time step, the potential energy of each bead must be computed; however, these calculations are independent and can be performed in parallel. We modified the Sander program to act as a driver program [34], whereby multiple instances of Sander can be launched, and communication between i-Pi and the pool of driver programs occurred through a network socket interface.

We recalculated the free energy profiles from PIMD umbrella sampling. The profiles generated from classical and path integral dynamics were compared to explore how nuclear quantum effects change the free energy predictions. The PIMD dynamics was propagated with 6 beads (replicas) and a 0.25 fs time step, and the temperature was maintained at 298 K using the PIGLET quantum thermostat [102,104], whose parameters were taken from the GLE4MD website [105,106]. The parameters were chosen to reproduce the quantum fluctuations at 298 K and span a range of frequencies up to 4142 cm^−1^. The PIMD umbrella sampling was performed with 32 windows using 200 kcal mol^−1^ Å^−2^ force constants. The windows uniformly discretized the optimized path obtained from classical dynamics. The PIMD restraint potentials were applied to the centroid positions by making use of the i-PI interface with PLUMED [107,108]. Each window was sampled for 2.5 ps, and the sampling was repeated 4 times with different thermostat random number seeds. The reaction coordinate values were calculated from the centroid positions, and the free energy surface was generated from MBAR analysis of the aggregate sampling obtained from the 4 trials of the 3 reaction steps.

Following Ref. [40], the interpretation of the PIMD results was aided by tracking the “degree of expansion” of the transferring proton ring polymer, $\Delta |r_{H}|$.
(1)$$\Delta |r_{H}|=\frac{1}{P}\sum_{n=1}^{P}\left|r_{H,n}-u_{H,c}\right|$$
where $r_{H,n}$ is the position vector of bead *n*, $u_{H,c}$ is the centroid position vector, and *P* is the number of beads. The degree of expansion is simply the average bead–centroid distance for the proton involved in the tautomeric reaction.

## 4. Conclusions

This work presents free energy simulation results of GT mispair tautomerization reactions in B-DNA using semiempirical QM/MM, *ab initio* QM/MM, and a new QM/MM-ΔMLP model that was specifically trained for this application. We demonstrate that short (2.5 ps) sampling of the reaction path discretized into 32 string images/umbrella windows leads to large deviations (up to 3.4 kcal/mol) in the calculated transition state barriers. The QM/MM-ΔMLP model enables greater sampling, which in turn leads to higher-precision free energy estimates, as well as offering the capability to include nuclear quantum effects through path integral molecular dynamics simulation. PIMD simulations produced barriers for the GT*→G*T interconversion that were significantly lower than those calculated from classical MD. Collectively, the results suggest that there is a rapid equilibrium for GT*⇌G*T but a relatively large barrier for the tautomerization of the wGT wobble pair to form either GT* or G*T tautomer states.

## Figures and Tables

**Figure 1 molecules-29-02703-f001:**
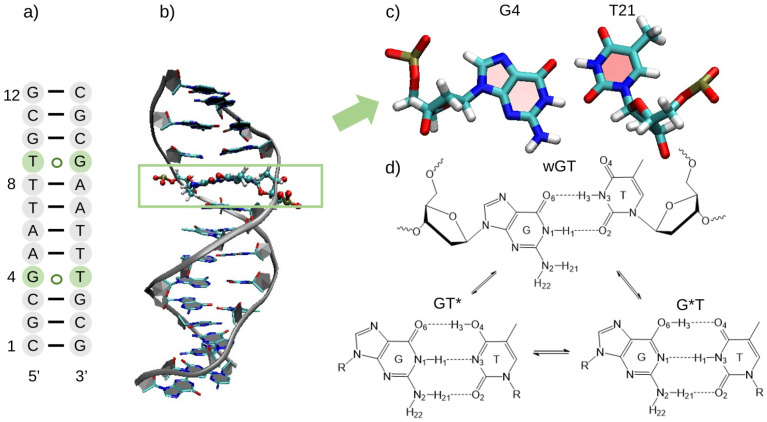
(**a**) B-DNA secondary structure with 2 G-T wobble mispairs highlighted in green; (**b**) B-DNA structure examined in this work prepared from the crystal structure [27] (PDB ID: 113D); (**c**) structure of the G-T wobble mispair residue; (**d**) schematic of the general reaction.

**Figure 2 molecules-29-02703-f002:**
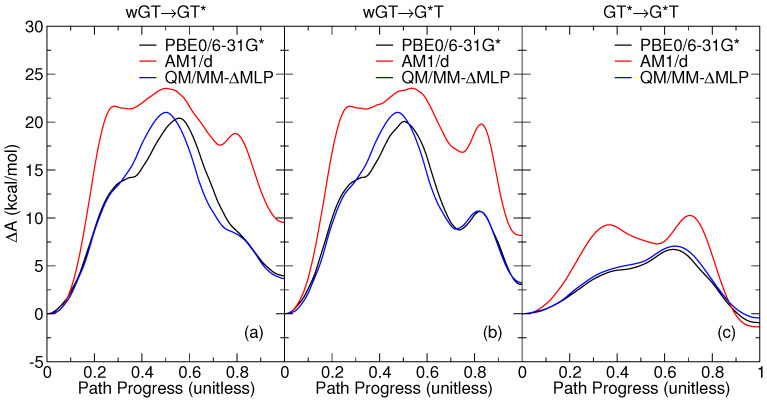
Classical free energy profiles for guanine–thymine mispair tautomerization reactions using different *ab initio* and semiempirical QM/MM and QM/MM-ΔMLP models described in the text: (**a**) wGT→GT*; (**b**) wGT→G*T; (**c**) GT*→G*T. Here “wGT” indicates a G-T wobble pair, and G* and T* are nonstandard (enol) tautomer states of G and T, respectively, as indicated in the figure and described in the text.

**Figure 3 molecules-29-02703-f003:**
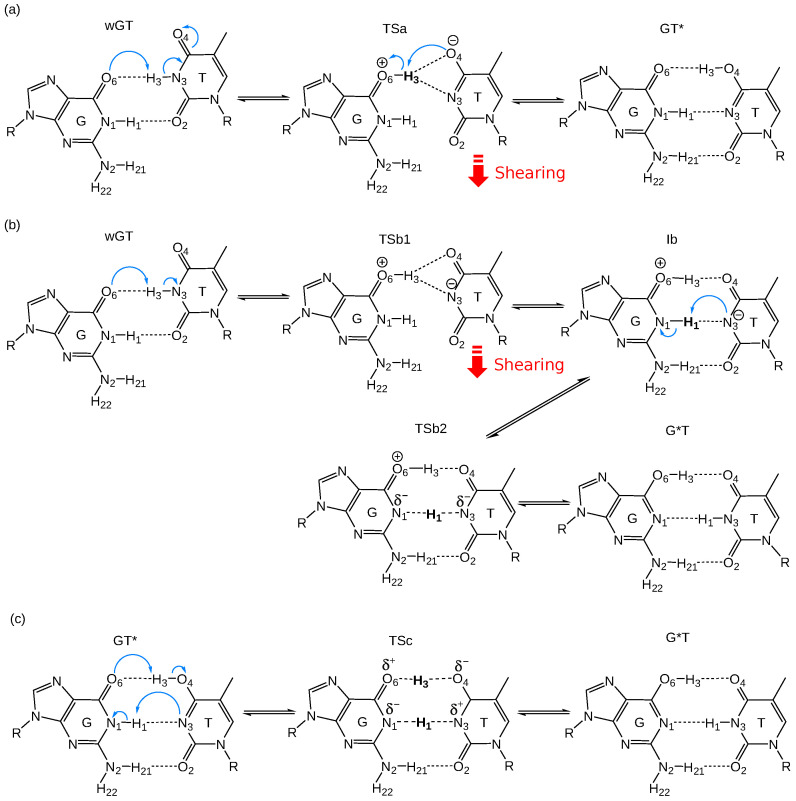
Schematic of guanine–thymine mispair tautomerization reaction mechanisms using QM/MM-ΔMLP models described in the text: (**a**) wGT→GT*; (**b**) wGT→G*T; (**c**) GT*→G*T. Here, “wGT” indicates a G-T wobble pair, and G* and T* are nonstandard (enol) tautomer states of G and T, respectively. Blue arrows represent electron transfer direction, and red motion arrows represent shearing direction of the base pairs [39].

**Figure 4 molecules-29-02703-f004:**
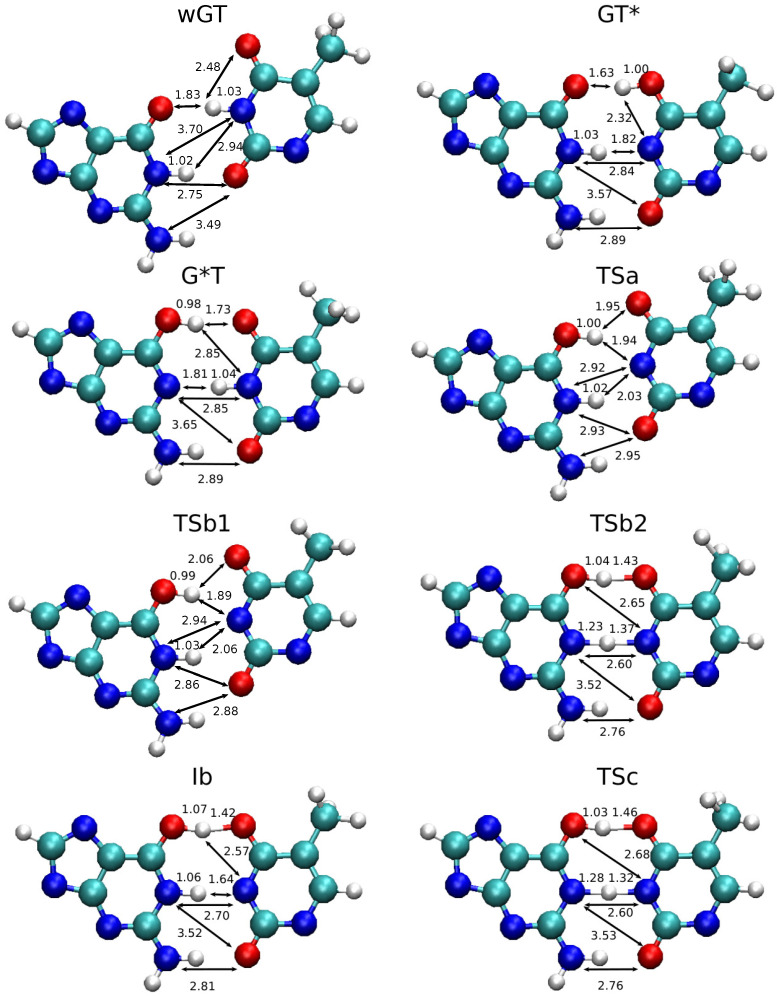
Representative nucleobase structures in guanine–thymine mispair tautomerization reactions using QM/MM-ΔMLP models described in the text and illustrated schematically in Figure 3. Here, “wGT” indicates a G-T wobble pair, and G* and T* are nonstandard (enol) tautomer states of G and T, respectively. TSa, TSb1, TSb2, and TSc are the transition states, and Ib is an intermediate state presented in Figure 2 and Figure 3. The structures were obtained by averaging the trajectory by Cpptraj [35], and the unit of distances shown here is Å. Other parts of the system, including sugar and phosphate groups, are not shown.

**Figure 5 molecules-29-02703-f005:**
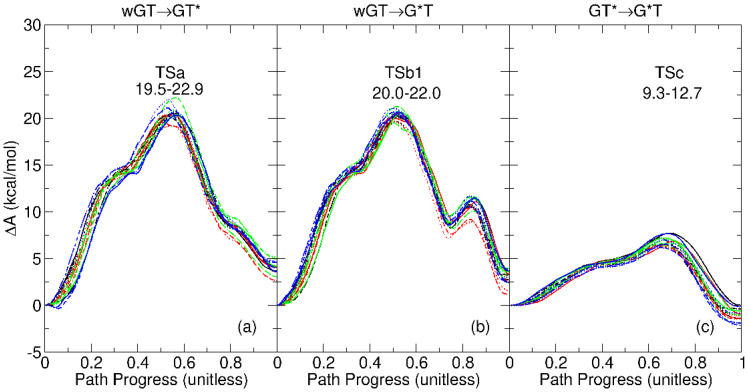
Classical free energy profiles for guanine–thymine mispair tautomerization reactions using *ab initio* QM/MM plotted with independent trials and analyzed with different 2.5 ps time blocks: (**a**) wGT→GT*; (**b**) wGT→G*T; (**c**) GT*→G*T. Here, “wGT” indicates a G-T wobble pair, and G* and T* are nonstandard (enol) tautomer states of G and T, respectively, as indicated in the figure and described in the text. Nonoverlapping 2.5 ps time blocks were constructed by uniformly dividing the 4 independent trials of 10 ps final path refinement sampling (40 ps cumulatively) into 16 2.5-ps blocks that were each analyzed independently. Different colors in the plots refer to different trials, and different line styles refer to different blocks. The ranges of rate-controlling transition state (TS) free energy barriers are given below the TS label.

**Figure 6 molecules-29-02703-f006:**
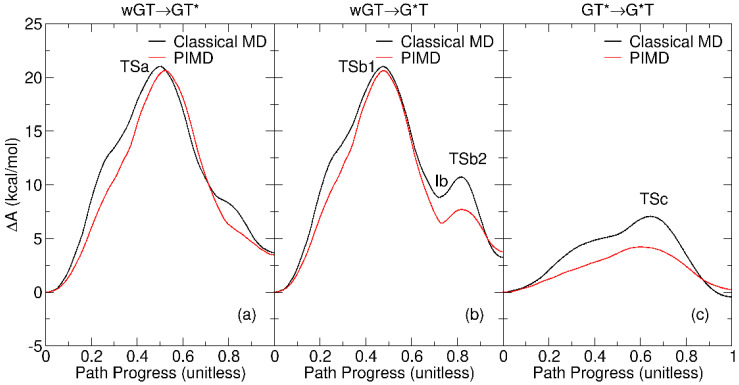
Classical and path integral molecular dynamics free energy profiles for guanine–thymine mispair tautomerization reactions using QM/MM-ΔMLP models described in the text: (**a**) wGT→GT*; (**b**) wGT→G*T; (**c**) GT*→G*T. Here, “wGT” indicates a G-T wobble pair, and G* and T* are nonstandard (enol) tautomer states of G and T, respectively. Profiles from classical and PIMD are shown as black and red lines, respectively.

**Figure 7 molecules-29-02703-f007:**
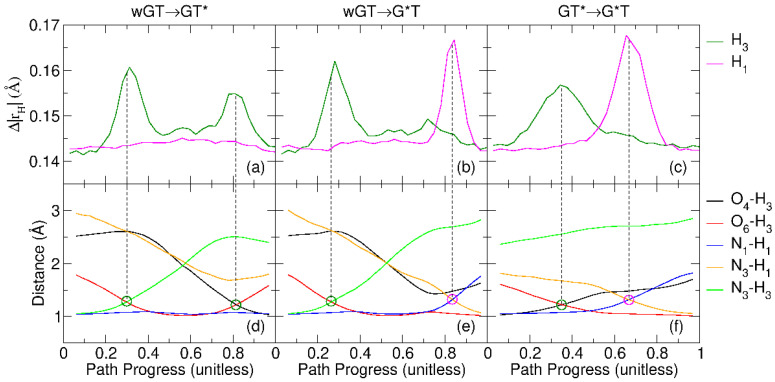
The atom distances and the proton ring polymer degree of expansion ($\Delta |r_{H}|$, Equation (Equation 1)) for guanine–thymine mispair tautomerization reactions using QM/MM-ΔMLP models described in the text: (**a**) wGT→GT* (**a**,**d**); (**b**) wGT→G*T (**b**,**e**); (**c**) GT*→G*T (**c**,**f**). Here, “wGT” indicates a G-T wobble pair, and G* and T* are nonstandard tautomer states of G and T, respectively. The proton ring polymer degree of expansion and the atom distances are shown as top and bottom panels, respectively. The notations of atoms are the same as in Figure 3.

## Data Availability

The original contributions presented in the study are included in the article/Appendix A; further inquiries can be directed to the corresponding author/s.

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
