# Peer review of "Electronic and Nuclear Quantum Effects on Proton Transfer Reactions of Guanine–Thymine (G-T) Mispairs Using Combined Quantum Mechanical/Molecular Mechanical and Machine Learning Potentials"

_molecules, 2024, doi:10.3390/molecules29112703_

Round 1

Reviewer 1 Report

Comments and Suggestions for Authors

In the article, Tao and co-workers present theoretical insights into a B-DNA system's G-T wobble misincorporation mechanism using ab initio and machine learning quantum mechanical/molecular mechanical methods. The general idea is that a proton must shift location for the G-T mispair to be included in the DNA, and the bases must slide against each other to form a Watson crick pairing via rearrangement of the hydrogen bonds.

The authors use a combination of ab initio and machine learning approaches to accelerate path sampling. They highlight that the system requires extensive sampling to determine precise energy barriers accurately. Furthermore, the authors utilize a path integral molecular dynamics formalism to describe the electronic and nuclear quantum effects on proton transfer.

The study concludes that including nuclear quantum effects results in only a modest change in the mechanistic pathway of the main G-T mispair reaction. However, the nuclear quantum effects caused a considerable reduction in the free energy barrier for the tautomeric reaction.

commend the authors for their work; the paper is interesting, and the data is generally well-presented and accessible. The level is appropriate to the journal's readership. Overall, the methodology is correct for answering the question; however, some aspects of its presentation are unclear. have split the recommendations into major and minor comments outlined below. 

Clarifications

Why is the reactant or product for the GT*->G*T reaction not zero for any figures?

The red arrows in Figure 3 show the shearing direction and do not point to the next row of figures?

Is the atom labeling in Figure 7 the same throughout the text and the figures?

Major comments

The paper is significantly long compared to the novelty of the result. Consequently, the authors should consider shortening sections of it to improve reader engagement and retention. This can be done without eroding the paper's substructure. suggest moving some sections of the paper to supplementary information so that the general readership can get the core findings, and additional info can still be found in the supplementary file. For example, some of the background in Section 1 could be moved to the supplementary file without undermining the arguments.

Could the authors comment on whether including the polymerase, similar to Li et al. (ref 18), would significantly modify the mechanistic pathway? Would this impact the nuclear quantum effects? Furthermore, the authors should provide a quantitative analysis of the change in rates and populations the PIMD would result in. 

While there is evidence supporting the conclusion, the discussion at the end could be further expanded to tie it back to the biochemical outlook and results. Discussing what this implies for the bio side of things would be good. It is not clear what the outcome suggests.

The use of ML to accelerate the path sampling convergence without loss of accuracy is a novel result that would be of widespread interest to the general readership. suggest modifying the text and the figures to make the benefits and pitfalls of this approach abundantly clear. leave this to the author's discretion, but for example, a suggestion would be to modify the caption or add labels to Figure 2 to demonstrate this. It would help to quantitatively compare the ML and the ab initio result in a table with further discussion. Furthermore, it is worth clarifying the dependence on the ab initio training data, such as the XC and the basis set.

Other authors have investigated the effects of nuclear quantum on this system. It is worth stating in the introduction the benefit of using PIMD over what has been done before. Furthermore, it could be better to highlight why nuclear effects are possible in this system, which is unclear from the abstract or introduction.

Minor suggestions

There are many acronyms in the abstract; one could remove PIMD.

Not to detract from Al-Hashimi et al.'s great work, but experimental evidence of this mechanism occurring was established before and might be worth raising.

Figure 4 is unclear and could be clarified regarding how it links back to the previous figures.

An intro figure is missing. It would be helpful to guide the reader to the problem at hand; Figure 1 helps but is quite far in.

There are some minor typesetting and grammar issues, but otherwise, it is very well presented—for example, line 77.

Overall, some issues must be addressed, and the authors can strengthen the paper by addressing my comments. look forward to seeing a final published version. 

Comments on the Quality of English Language

A few sentences might need cleaning up, but overall, the quality of written English is superb.

Reviewer 2 Report

Comments and Suggestions for Authors

Authors have studied the PMF and mechanism of tautomerization of GT pairs using PIMD and ML based protocols. The manuscript is well written and should be accepted after a few minor suggestions.

1.  Authors have shown in Fig 1, that the free energy profile  obtained from PBE functional calculations to be very different from that based on ML based protocols and AM1 calculations.  How do authors ensured that in the later calculations this actually got the opposite with highly accurate numbers in agreement with the QM method.

2. Moreover, the free energy curves are very different for ML and AM1 based calculations.  This means that several intermediates predicted using QM might not be observable in ML calculations.

3.  In Fig 6 (a), the difference between classical and PIMD is not discernable. Do the authors imply that this step which has hydrogen transfer/bonding no NQE or EQE? Do they cancel out each other?

4. I will like to see a paragraph where authors categorically mention each step reaction and describe the NQE and EQE and how much do they contribute to the overall free energy of the step.   

Comments on the Quality of English Language

Language is verbose and not easy to read.

Round 2

Reviewer 1 Report

Comments and Suggestions for Authors

I want to thank the authors for thoughtfully incorporating my suggestions. I now believe the paper is in a very good place. It is clear and concise, and I enjoyed reading the final version. Great work!